# Why Do Medical Time Series Models for Disease Detection Generalize Poorly to Unseen Subjects?

## Abstract

Models for disease detection in medical time series (MedTS) often excel on training subjects but fail to generalize to unseen subjects. In many disease detection datasets, each subject is associated with a single, fixed label, resulting in strong yet spurious correlations. Across EEG- and ECG-based disease detection, spurious identity correlations inflate performance in *subject-dependent* evaluations (shared subjects across train/test) but collapse under *subject-independent* splits with unseen test subjects. Our comparative experiments indicate that disease detection models often exploit the shortcut of patient identity, severely limiting their generalization to unseen subjects. These findings highlight the critical need for methods designed to mitigate subject identity as a spurious feature and reinforce the importance of subject-independent setup for clinically meaningful MedTS disease detection.

## 1 Introduction

Medical time series (MedTS) are specialized time series data representing continuous recordings of physiological signals from human subjects, such as EEG, ECG, fNIRS, and PPG (Liu et al., 2021; Esgalhado et al., 2021; Eastmond et al., 2022; Wang et al., 2024b; Badr et al., 2024). Beyond typical time series data, MedTS capture each individual's distinct physiological traits, from overt demographics to subtler attributes like anatomical structures and brain activity patterns, that could introduce subject-specific information not necessarily tied to the task label (Wang et al., 2024c). In this work, we refer to features unique to each subject but unrelated to the label as **identity features** that are *subject-specific*, and we refer to features that are shared across individuals and pertinent to the label as **task** or **disease features** that are *subject-invariant*. While a model's primary objective is to learn task features for classification, we assume the presence of strong identity features can distort the learning process. Our experiments across multiple EEG and ECG datasets reveal that identity features dominate. For example, on the ADFTD dataset (Miltiadous et al., 2023b), a leading baseline achieves only 52.37% accuracy in a 3-way disease classification on new subjects but achieves 98.78% accuracy in an 88-way subject discrimination task. This disparity raises an important concern: Is the model truly learning disease features, or is it relying on salient identity features despite being trained for disease detection?

To explore this question, we begin by considering whether a naive shortcut, one that relies solely on identity features, could solve the task. In a scenario where each subject has only one fixed label, a model that perfectly distinguishes subjects would also achieve 100% accuracy on that task by memorizing a direct mapping from subject ID to label. However, if each subject's task label takes on $n$ different categories, an identity-only model could reach at best $1/n$ accuracy without using any task-relevant information. In other words, the performance of an optimal identity-only model depends on the strength of the one-way *spurious correlation* from subject ID to the task label. This observation motivates us to categorize MedTS datasets based on whether each subject's label is fixed or dynamic. For completeness, we introduce three dataset types: **Type-I**, **Type-II**, and **Type-III** MedTS corresponding to single-subject multi-class, multi-subject multi-class, and multi-subject single-class, respectively. (Figure 1a; see Section 2.1 for details). Most disease detection datasets fall under Type-III MedTS: they comprise multiple subjects, each with a single fixed condition (healthy or diseased), since collecting both healthy and diseased signals (e.g., Alzheimer's Disease) from the same individual is typically impractical.

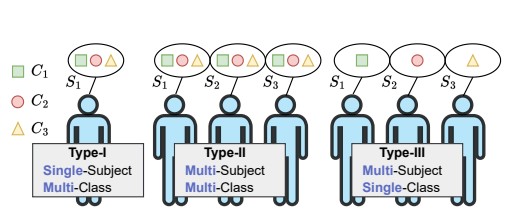 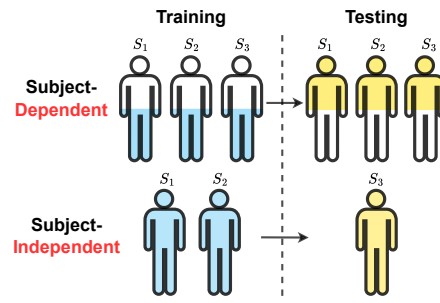

(a) Types of MedTS Datasets  (b) Subject-Dependent and -Independent Setups

Figure 1: **MedTS types and evaluation protocols.** (a) Three categories of MedTS datasets. S and C denote subject and class, respectively. (b) Illustration the differences between the two main setups Wang et al. (2024b): subject-dependent and subject-independent.

The potential shortcut via identity also underscores the importance of proper evaluation protocols. In non-MedTS datasets, randomly splitting samples into training and test sets is often sufficient. However, *in MedTS datasets, such random splitting can place samples from the same subject in both sets, allowing a model that learns an identity-to-label shortcut to apply it unchanged at test time.* To nullify this shortcut, the test set must include only *unseen* subjects, achieved by splitting the dataset at the subject level rather than the sample level. We refer to this as the **subject-independent** setup, whereas a sample-level split as the **subject-dependent** setup (Figure 1b).

Subject-independent splitting offers two major benefits over subject-dependent splitting. First, it reflects a realistic clinical scenario where a model trained on data from a small cohort must be applied to a wider population of completely new individuals. Second, and more critically, whereas subject-dependent evaluations condone identity-based shortcut learning, subject-independent evaluations *expose* it. While an increasing number of studies have recognized the value of subject-independent protocols (Sherman et al., 2017; Gholamiangonabadi et al., 2020; Kunjan et al., 2021; Wang et al., 2024b; Tu et al., 2024), the majority of research in AI for medical time-series still use subject-dependent evaluations without an explicit discussion of their limitations. We suspect this stems from a general lack of domain-specific expertise in medical data evaluation among many AI researchers. A brief survey of published Alzheimer's Disease detection papers in the last year alone (Nour et al., 2024; Puri et al., 2024; Bravo-Ortiz et al., 2024; Tran et al., 2024; Kachare et al., 2024; Sen et al., 2024) shows that subject-dependent splits are still prevalent, often yielding deceptively high metrics (over 90% accuracy).

**This paper systematically analyzes how identity-based shortcuts in Type-III MedTS datasets inflate model performance in subject-dependent evaluations, demonstrating the necessity of subject-independent protocols for reliable disease detection**. In type-III datasets, each subject's label is fixed. Hence, subject identity fully overlaps with the disease label, resulting in a strong spurious correlation. We demonstrate empirically that under a *subject-dependent* setup, Models can attain high disease detection accuracy by exploiting subject identity as a shortcut rather than genuinely learning disease-relevant features (see Table 4). By contrast, once the evaluation is conducted in a *subject-independent* manner, where the model encounters unseen subjects, this overreliance is revealed through a sharp performance drop (49% F1 in ADFTD, 28% F1 in PTB-XL, see Table 4).

To thoroughly investigate this phenomenon, we conduct experiments on four real-world MedTS datasets (2 EEG and 2 ECG). A key component of our methodology is a *label-shuffling* procedure, where each subject is randomly assigned a new label at the subject level. This artificial labeling severs the genuine link between disease features and labels, leaving only subject identity as a possible basis for classification. Models still perform remarkably good under subject-dependent splits even after labels are shuffled, indicating that identity-based features alone can be used to predict the label.

For instance, using Medformer (Wang et al., 2024c) on the ADFTD and PTB datasets, F1 scores drop only marginally when disease-related features are eliminated (e.g., 97.56% to 97.12% on ADFTD), demonstrating that subject identity can suffice to classify labels in a subject-dependent scenario. Even on the larger PTB-XL dataset, the subject-dependent accuracy remains substantially above baseline,

yet it falls dramatically in subject-independent settings (e.g., from 88.02% down to 71.15% when disease features are removed, still higher than the 59.97% observed under a truly subject-independent protocol). These outcomes show that a *subject-dependent evaluation can vastly overestimate a model's disease-discriminative power by capturing spurious subject cues.*

Our findings appeal to researchers in the AI4health community to use subject-independent evaluations for Type-III MedTS disease detection. Subject-dependent splits not only inflate performance metrics but can be outright misleading, effectively measuring how well a model recognizes subjects rather than detects disease. Looking forward, more robust strategies—ranging from specialized data augmentations (Wang et al., 2024b) to novel network regularization or hybrid learning mechanisms—will be crucial for disentangling identity features from genuine disease indicators. We hope these results urge the community to adopt subject-independent protocols and inspire future work on building clinically meaningful and generalizable MedTS models.

## 2    Taxonomy of MedTS Datasets and Evaluation Setups

### 2.1    Types of Medical Time Series Dataset

We categorize MedTS datasets into three types below based on the number of subjects and classes:

**Single-Subject and Multi-Class (Type-I).** Type-I MedTS is typically used to develop models tailored to a specific subject. Examples include designing brain-computer interfaces for individuals with disabilities (Musk et al., 2019; Scherer et al., 2015) or building personalized health monitoring systems (Levett et al., 2023) using wearable devices. In these scenarios, the model focuses on learning from the time-varying classes generated by a single subject's data over time. In Type-I datasets, there is no spurious correlation between identity and task as there is only one subject

**Multi-Subject and Multi-Class (Type-II).** Type-II is used to design models that can adapt to multiple subjects with varying classes. For instance, mental state recognition (Zhang et al., 2018; Dai et al., 2019; Li et al., 2022; Jafari et al., 2023) or sleep state classification (Mousavi et al., 2019; Eldele et al., 2021) fall under this category. In this case, the MedTS samples from different subjects are associated with dynamically changing classes over time, meaning no fixed class is assigned to each subject. In Type-II datasets, there is a weak spurious correlation between identity and task.

**Multi-Subject and Single-Class (Type-III).** In Type-III, each subject is assigned a single class that remains consistent over time. For example, once a subject is diagnosed with Alzheimer's Disease during the data collection stage, this subject is always classified as an AD patient. This type of MedTS is often used as a low-cost diagnostic method to supplement or replace more traditional medical approaches (Wagner et al., 2020; van Dijk et al., 2022). It is important to clarify that the term "single-class" here refers to cases where the medical or physiological state of a subject remains fixed over time (or within a short period without significant change). For example, a patient diagnosed with Alzheimer's Disease (AD) will typically remain in that state for decades after the diagnosis. In Type-III datasets, there is a **100%** spurious correlation from identity to label.

### 2.2    MedTS Evaluation Setups

We categorize the commonly used evaluation setups in existing MedTS classification studies into two main types below (Figure 1b) (Seal et al., 2021; Wang et al., 2024b):**1) Subject-Dependent.** Samples from different subjects are randomly split into training, validation, and test sets, allowing the same subject's data to appear in all three sets. **2) Subject-Independent.** The training, validation, and test sets are split on a subject level, ensuring that samples from the same subject are exclusively included only in one set. More evaluation protocol details in Appendix B.

In evaluation setups, the inclusion of a *validation set* and *cross-validation* are optional but strongly recommended for real-world applications to ensure a comprehensive and reliable evaluation. Additionally, if the dataset lacks subject IDs, subject-independent splitting becomes infeasible, forcing the use of subject-dependent evaluation protocols.

Table 1: **Feature Components Utilization of Setups.** The table provides an overview of the feature components utilized by a trained model $f_\theta$ when classifying a given Type-III MedTS sample $x$ during evaluations across five experimental setups, along with their corresponding classification tasks. The setups include subject-dependent (Sub-Dep), subject-independent (Sub-Indep), subject-discrimination (Sub-Disc), random-label subject-dependent (R-Sub-Dep), and random-label subject-independent (R-Sub-Indep). There are three feature components considered: disease-related features $x^d$, subject-specific features $x^s$, and all other features $x^o$. The classification tasks involve predicting either the label $y$ or the subject ID $z$ of a sample.

| | Feature Components Utilization | | | Classification Tasks | |
| --- | --- | --- | --- | --- | --- |
| Setups | $x^d$ (Disease) | $x^i$ (Identity) | $x^o$ (Other) | $y$ (Label) | $z$ (ID) |
| Sub-Dep | ✓ | ✓ | ✓ | ○ | |
| Sub-Indep | ✓ | | ✓ | ○ | |
| Sub-Disc | | ✓ | ✓ | | ○ |
| R-Sub-Dep | | ✓ | ✓ | ○ | |
| R-Sub-Indep | | | ✓ | ○ | |

## 3 METHOD

This paper focuses on disease detection within the **Type-III** MedTS dataset, characterized by multiple subjects with a single class assigned per subject (section 2.1; Figure 1a).

In this section, we first introduce our notations and assumptions for describing the feature components of Type-III MedTS. Next, we analyze the subject-dependent and subject-independent setups in the context of feature components utilized in the Type-III MedTS classification tasks. Finally, we propose several new experimental setups based on these two setups to validate our assumptions systematically.

### 3.1 NOTATIONS AND ASSUMPTIONS

Given a Type-III MedTS sample $x \in \mathbb{R}^{T \times C}$, where $T$ denotes the number of timestamps and $C$ represents the number of channels, the sample is associated with a corresponding disease-related label $y \in \mathbb{R}^K$. Here, $K$ indicates the number of medically relevant classes, such as different disease types. Each sample is also assigned to a subject ID $z \in \mathbb{R}$, identifying the subject to which it belongs.

In time series forecasting tasks, researchers commonly decompose data into trend and seasonal feature components and design mechanisms to effectively leverage these components for improved representation learning (Wu et al., 2021; Fraikin et al., 2023; Wang et al., 2024a). Similarly, Type-III MedTS data also have several distinct feature components. We assume that a Type-III MedTS sample $x$ consists of three feature components: $x = x^d \otimes x^i \otimes x^o$ where $x^d$ represents **disease features**, $x^i$ denotes disease unrelated **identity features**, and $x^o$ includes **all other features**, such as noise, artifacts, and all task-irrelevant and identity-irrelevant features. In disease detection tasks, our goal is to train a model $f_\theta : x \to y$ that accurately maps a given sample $x$ to its corresponding disease-related label $y$. Ideally, the model $f_\theta$ should rely solely on $x^d$ and be entirely independent of identity features $x^i$ when predicting the label $y$.

Given the unique nature of Type-III MedTS data, where *samples with the same subject ID $z$ always share the same label $y$*, there is a risk that the model $f_\theta$ might exploit identity features $x^i$ as a shortcut for predicting $y$, rather than learning the disease-related features $x^d$. Although some existing studies have attempted to address this issue and propose methods to tackle it, effectively disentangling $x^d$ and $x^i$ from $x$ remains an open challenge (Zhang et al., 2020; Yang et al., 2022; Wang et al., 2024c).

To summarize, we assume that *identity features $x^i$ exist in each sample $x$ within the Type-III MedTS dataset, and it is possible for the trained model $f_\theta$ to utilize $x^i$ as a shortcut for classifying the label $y$. This could result in deceptively high performance in subject-dependent evaluations while the model learns little to nothing about the disease-related features $x^d$.*

### 3.2 SUBJECT-DEPENDENT VS INDEPENDENT

We analyze the differences of two setups from the perspective of feature components used when classifying a Type-III MedTS sample $x$. Their feature components utilization are presented in table 1.

Table 2: The information for the processed datasets. The table shows the number of subjects, samples, classes, channels, sampling rate, sample timestamps, type of MedTS, and file size. Here, #-Timestamps indicates the number of timestamps per sample.

| Datasets | #-Subject | #-Sample | #-Class | #-Channel | #-Timestamps | Sampling Rate | MedTS-Type | File Size |
|---|---|---|---|---|---|---|---|---|
| ADFTD | 88 | 69,752 | 3 | 19 | 256 | 256Hz | Type-III | 2.52GB |
| TDBrain | 72 | 6,240 | 2 | 33 | 256 | 256Hz | Type-III | 571MB |
| PTB | 198 | 64,356 | 2 | 15 | 300 | 250Hz | Type-III | 2.15GB |
| PTB-XL | 17,596 | 191,400 | 5 | 12 | 250 | 250Hz | Type-III | 4.28GB |

**a) Subject-Dependent.** The subject-dependent setup disregards the subject ID, treating the Type-III MedTS classification problem similarly to general time series classification. Samples from different subjects within the dataset are shuffled, mixed, and randomly divided into training, validation, and test sets. Samples sharing the same subject ID can be present inclusively in these three sets.

In this scenario, for a given sample $x$ in the validation/test set, the trained model $f_\theta$ could leverage both its disease-related features $x^d$ and identity features $x^i$ to predict the label $y$. Consequently, this setup can lead to an overestimation of the model's performance, as it can exploit identity information rather than generalizing based on disease-related features alone.

**b) Subject-Independent.** The subject-independent setup considers the effect of identity features and closely simulates real-world applications. In this setup, the training, validation, and test sets are split based on subjects rather than individual samples, ensuring that all samples from a particular subject (i.e., with the same subject ID) are exclusively included in one of the three sets.

In this scenario, for any given sample $x$ in the validation/test set, the trained model $f_\theta$ can only leverage the disease-related features $x^d$ to predict the label $y$, as no samples with the same subject ID $z$ are present in the training set. This setup aligns with real-world applications of disease diagnosis using Type-III MedTS, where a label is assigned to an entire subject, and it's impractical to have access to labeled samples from that subject in advance during training.

However, this setup is challenging and can exhibit poor results. Firstly, while the subject independent setup prevents model $f_\theta$ from exploiting identity features $x^i$ as a shortcut for classification during validation and testing, it remains possible for the model to rely on identity features during the training stage, impeding its ability to learn task-related subject-invariant features. Secondly, from a domain generalization perspective, different subjects can be considered distinct domains (Yang et al., 2022), making it challenging to efficiently extract disease-related features that generalize well across unseen domains (i.e., subjects in the validation/test sets). Currently, there are no effective methods to fully disentangle identity and disease-related features. This remains an open challenge in the field.

## 3.3 CUSTOM EXPERIMENTAL SETUPS

To demonstrate that identity features are universally present in Type-III MedTS datasets, we first introduce the subject-discrimination setup to identify these features. Following this, we design two additional setups, including random-label subject-dependent and random-label subject-independent, to intentionally sever the connection between disease features and the label. These setups allow us to validate our assumption that models trained under the subject-dependent setup heavily rely on identity features as a shortcut for label classification. The feature components utilized and classification tasks for three new setups are present in table 1. The experimental results in Sections 4.2-4.4.

**a) Subject-Discrimination.** The disease label $y$ is assigned by medical experts from data, ensuring that disease-related features $x^d$ are present in the MedTS samples for each subject. However, there is no guarantee that identity features $x^i$ exist, although researchers usually assume they exist. To investigate the presence of identity features $x^i$ for samples in each subject, we design this setup to prove the existence of identity features by trying to classify the subject ID $z$ of samples.

In this experiment, we randomly split all samples into training, validation, and test sets. We train a model $f_\theta$ to predict the subject ID $z$ of a given sample $x$. Since the number of subjects is much larger than the number of disease labels, the model cannot rely on disease-related features $x^d$ as a shortcut for subject ID classification. Therefore, if the model performs well in classifying the subject ID $z$, it strongly indicates the existence of identity features $x^i$ within the dataset.

**b) Random-Label Subject-Dependent.** Previous research has consistently shown that models evaluated under the subject-dependent setup generally achieve much higher performance compared to the subject-independent setup (Nath et al., 2020; Arif et al., 2024; Wang et al., 2024c). While it is widely assumed that the model under the subject-dependent setup exploits identity features $x^i$ as a shortcut for classifying the label $y$, the extent of the model's reliance on $x^i$ and whether it still learns disease-related features $x^d$ remains unclear. To investigate the clear influence of identity features $x^i$ on model performance, we introduce the random-label subject-dependent setup.

This setup differs slightly from the original subject-dependent setup. We randomly shuffle the label of each subject before splitting the samples into training, validation, and test sets. This operation disrupts the relationship between disease-related features $x^d$ and the label $y$, effectively preventing the model $f_\theta$ from leveraging $x^d$ for prediction.

As a result, in the random-label subject-dependent setup, the trained model $f_\theta$ can rely solely on identity features $x^i$ to predict the label $y$ for any given sample $x$. By comparing the performance of the random-label subject-dependent setup with the original subject-dependent setup, we can directly assess the extent to which the model relies on disease feature $x^d$ relative to identity feature $x^i$.

**c) Random-Label Subject-Independent.** Similar to the random-label subject-dependent setup, we apply the same process in the subject-independent setup, where we randomly shuffle the label of each subject while retaining their subject IDs before splitting the data into training, validation, and test sets. This ensures that the model $f_\theta$ cannot learn any disease-related features $x^d$.

The random-label subject-independent setup serves as a control for the random-label subject-dependent setup, enabling us to isolate the potential impact of other features $x^o$. In this setup, the model is unable to utilize either disease-related features $x^d$ or identity features $x^i$ to classify the label $y$; instead, it can only rely on other features $x^o$. In other words, if the other features $x^o$ are independent of the label $y$, the performance under the random-label subject-independent setup should be completely random.

## 4 EXPERIMENTS

To make the experimental scenarios robust and pervasive, we train on four different models: Multi-Layer Perception (MLP) (Popescu et al., 2009), Temporal Convolutional Neural Networks (TCN) (Bai et al., 2018), vanilla Transformer (Vaswani et al., 2017), and Medformer (Wang et al., 2024c), to cover different architectures from MLP, CNN, to transformer. The implementation details are presented in Appendix C. We evaluate four MedTS datasets, including two EEG datasets, ADTFD (Miltiadous et al., 2023b), TDBrain (van Dijk et al., 2022), and two ECG datasets, PTB (PhysioBank, 2000) and PTB-XL (Wagner et al., 2020). The data pre-processing are presented in Appendix A. The information of processed datasets is listed in Table 2. We employ two key evaluation metrics: accuracy and F1 score (macro-averaged). The training process is conducted with five random seeds (41-45) to compute the mean and standard deviation of the metrics. All experiments are run on an NVIDIA RTX 4090 GPU and a server with 4 RTX A5000 GPUs.

Table 3: The results of disease detection datasets evaluated under the subject-discrimination(Sub-Disc) setup are summarized in the table. This setup aims to verify the existence of subject-specific features in the datasets. The high performance observed across all four datasets supports this assumption, indicating that subject-specific features $x^i$ are present.

| Models | | MLP | | TCN | | Transformer | | Medformer | |
|---|---|---|---|---|---|---|---|---|---|
| Datasets | Setups | Accuracy | F1 Score | Accuracy | F1 Score | Accuracy | F1 Score | Accuracy | F1 Score |
| ADFTD (88-Subjects) | Sub-Disc | 30.89±0.66 | 30.90±0.61 | 86.31±1.60 | 86.10±1.59 | 98.15±0.49 | 98.00±0.52 | **98.78±0.16** | **98.72±0.17** |
| TDBrain (72-Subjects) | Sub-Disc | 29.78±0.73 | 28.83±0.73 | 58.23±4.03 | 56.92±4.42 | 80.93±1.78 | 80.14±1.83 | **86.93±1.50** | **86.56±1.58** |
| PTB (198-Subjects) | Sub-Disc | 99.16±0.02 | 99.21±0.05 | **99.68±0.03** | **99.68±0.03** | 99.54±0.03 | 99.60±0.04 | 99.64±0.04 | 99.66±0.04 |
| PTB-XL (17,596-Subjects) | Sub-Disc | 5.32±0.13 | 4.68±0.11 | **84.04±2.39** | **84.35±2.18** | 69.92±1.51 | 68.01±1.67 | 78.27±0.50 | 76.64±0.53 |

## 4.1 RESULTS OF SUBJECT-DEPENDENT AND SUBJECT-INDEPENDENT

**Setup.** We begin by comparing the results between the subject-dependent and subject-independent setups. In the subject-dependent setup, the training, validation, and test sets are split based on individual samples, allowing samples with the same subject ID to be present in all three sets simultaneously. In contrast, the subject-independent setup splits these sets based on subjects, ensuring that samples with the same subject ID are exclusively included in only one of the three sets. As discussed in Section 3.2, for a given sample $x$, a trained model $f_\theta$ following the subject-dependent setup can leverage both disease-related features $x^d$ and identity features $x^i$ to classify the label $y$. However, in the subject-independent setup, the trained model $f_\theta$ can rely solely on the disease-related features $x^d$ during evaluation. By comparing the results between these two setups, we can gain insights into the extent to which identity features influence the model's performance.

**Results.** The results are summarized in Table 4, where "Sub-Dep" and "Sub-Indep" refer to the subject-dependent and subject-independent setups, respectively. For each dataset, we highlight in bold the best-performing result across the four methods within each setup. Across all datasets, it's evident that the subject-dependent setup consistently outperforms the subject-independent setup, regardless of the training method used. The performance gap between the subject-dependent and subject-independent setups varies across different datasets. For instance, when using the Medformer model, the F1 score difference between two setups is approximately 50% F1 score for the ADFTD dataset, but only around 13% F1 score for the TDBrain dataset. This variation suggests that the ratio of disease-related features $x^d$ to identity features $x^i$ differs across datasets.

## 4.2 RESULTS OF SUBJECT-DISCRIMINATION

**Setup.** The subject-discrimination setup is similar to the subject-dependent setup in that we randomly shuffle, mix, and split all samples into training, validation, and test sets, allowing samples with the same subject ID to be present in all three sets simultaneously. However, the classification task in this setup is shifted from predicting the label $y$ to identifying the subject ID $z$. The primary goal of this setup is to confirm the existence of identity features $x^i$ within a dataset and to assess their strength based on the model's classification performance. As discussed in Section 3.3, when validating or testing a given sample $x$, the trained model $f_\theta$ is unable to rely on disease-related features $x^d$ as shortcuts for subject ID $z$ classification, since there is typically a one-to-many relationship between disease label $y$ and subject ID $z$. In rare cases where a dataset of two subjects has opposite labels, the model $f_\theta$ might theoretically use disease-related features $x^d$ as a shortcut to classify the subject ID $z$. However, this scenario does not apply to the datasets used in this paper, where the number of subjects far exceeds the number of labels.

**Results.** The results of the subject-discrimination setup are presented in Table 3. The highest F1 in all models is 98. 72% in ADFTD, 86. 56% in TDBrain, 99. 68% in PTB, and 84. 35% in PTB-XL. The performance for ADFTD and PTB is remarkably high, approaching an F1 score of 100%, indicating that these two datasets has very strong identity features. Even on the large PTB-XL dataset (17,596 subjects) the best model still achieved an F1 above 80%. Overall, these results provide compelling evidence for the existence of identity features across all four datasets.

## 4.3 RESULTS OF RANDOM-LABEL SUBJECT-DEPENDENT

**Setup.** The random-label subject-dependent setup builds upon the subject-dependent setup with an additional step. Before shuffling and splitting all samples into training, validation, and test sets, we first randomly assign a new label to each subject. For instance, in the ADFTD dataset, which has three classes (0, 1, and 2) representing different medical conditions, we randomly assign each subject a new label from these three classes, replacing their true label. The purpose of this setup is to deliberately mask the disease-related features in the dataset, ensuring that the trained model $f_\theta$ theoretically cannot learn any disease-related features during training. As a result, the model can only rely on identity features to classify the label $y$ of a given sample $x$ during validation and test. The performance drop observed in this setup compared to the original subject-dependent setup will indicate the extent to which the model learned disease-related features in the subject-dependent setup.

**Results.** The results of the random-label subject-dependent setup are presented in Table 4. For comparison, we also include the results from the original subject-dependent and subject-discrimination

Table 4: The results of disease detection MedTS datasets evaluated under subject-dependent(Sub-Dep), random-label subject-dependent(R-Sub-Dep), and subject-independent setups. The negligible performance drop from the subject-dependent to the random-label subject-dependent setup of ADFTD and PTB indicates that the model learns almost nothing about disease-related features $x_d$ during training under the subject-dependent setup for these two datasets.

| Models | | MLP | | TCN | | Transformer | | Medformer | |
|---|---|---|---|---|---|---|---|---|---|
| Datasets | Setups | Accuracy | F1 Score | Accuracy | F1 Score | Accuracy | F1 Score | Accuracy | F1 Score |
| ADFTD (3-Classes) | Sub-Dep | $58.66_{\pm0.61}$ | $55.48_{\pm0.44}$ | $81.42_{\pm0.49}$ | $80.25_{\pm0.54}$ | $97.00_{\pm0.43}$ | $96.86_{\pm0.44}$ | $\mathbf{97.66}_{\pm\mathbf{0.76}}$ | $\mathbf{97.56}_{\pm\mathbf{0.79}}$ |
| | R-Sub-Dep | $46.33_{\pm1.48}$ | $45.88_{\pm1.41}$ | $75.16_{\pm1.68}$ | $75.00_{\pm1.82}$ | $96.99_{\pm0.29}$ | $96.97_{\pm0.29}$ | $\mathbf{97.14}_{\pm\mathbf{0.74}}$ | $\mathbf{97.12}_{\pm\mathbf{0.74}}$ |
| | Sub-Indep | $49.10_{\pm1.01}$ | $43.78_{\pm0.28}$ | $50.46_{\pm1.35}$ | $47.32_{\pm1.27}$ | $50.47_{\pm2.14}$ | $48.09_{\pm1.59}$ | $\mathbf{52.37}_{\pm\mathbf{1.51}}$ | $\mathbf{48.72}_{\pm\mathbf{1.18}}$ |
| TDBrain (2-Classes) | Sub-Dep | $81.93_{\pm0.26}$ | $80.16_{\pm0.29}$ | $97.32_{\pm0.28}$ | $97.16_{\pm0.30}$ | $\mathbf{97.60}_{\pm\mathbf{0.19}}$ | $\mathbf{97.45}_{\pm\mathbf{0.21}}$ | $96.70_{\pm0.42}$ | $96.51_{\pm0.45}$ |
| | R-Sub-Dep | $63.40_{\pm2.22}$ | $62.42_{\pm2.29}$ | $\mathbf{89.88}_{\pm\mathbf{2.26}}$ | $\mathbf{89.71}_{\pm\mathbf{2.21}}$ | $86.88_{\pm1.26}$ | $86.65_{\pm1.35}$ | $86.17_{\pm2.71}$ | $85.95_{\pm2.72}$ |
| | Sub-Indep | $69.42_{\pm0.64}$ | $69.37_{\pm0.64}$ | $83.98_{\pm2.31}$ | $83.93_{\pm2.35}$ | $\mathbf{86.58}_{\pm\mathbf{0.76}}$ | $\mathbf{86.52}_{\pm\mathbf{0.79}}$ | $83.92_{\pm1.01}$ | $83.69_{\pm1.09}$ |
| PTB (2-Classes) | Sub-Dep | $99.80_{\pm0.02}$ | $99.63_{\pm0.03}$ | $\mathbf{99.95}_{\pm\mathbf{0.01}}$ | $\mathbf{99.91}_{\pm\mathbf{0.03}}$ | $99.92_{\pm0.02}$ | $99.86_{\pm0.04}$ | $99.94_{\pm0.02}$ | $99.90_{\pm0.05}$ |
| | R-Sub-Dep | $99.44_{\pm0.02}$ | $99.44_{\pm0.03}$ | $\mathbf{99.83}_{\pm\mathbf{0.03}}$ | $\mathbf{99.83}_{\pm\mathbf{0.03}}$ | $99.64_{\pm0.11}$ | $99.64_{\pm0.11}$ | $99.69_{\pm0.04}$ | $99.69_{\pm0.04}$ |
| | Sub-Indep | $77.76_{\pm0.46}$ | $70.02_{\pm0.60}$ | $\mathbf{83.97}_{\pm\mathbf{2.26}}$ | $\mathbf{78.99}_{\pm\mathbf{3.44}}$ | $77.37_{\pm1.02}$ | $68.47_{\pm2.19}$ | $77.86_{\pm1.64}$ | $69.93_{\pm2.69}$ |
| PTB-XL (5-Classes) | Sub-Dep | $66.99_{\pm0.10}$ | $52.98_{\pm0.24}$ | $88.61_{\pm0.42}$ | $85.21_{\pm0.73}$ | $87.86_{\pm0.32}$ | $84.50_{\pm0.41}$ | $\mathbf{90.48}_{\pm\mathbf{0.24}}$ | $\mathbf{88.02}_{\pm\mathbf{0.33}}$ |
| | R-Sub-Dep | $21.62_{\pm0.15}$ | $21.58_{\pm0.13}$ | $\mathbf{72.55}_{\pm\mathbf{0.39}}$ | $\mathbf{72.54}_{\pm\mathbf{0.39}}$ | $62.53_{\pm2.07}$ | $62.52_{\pm2.07}$ | $71.16_{\pm1.58}$ | $71.15_{\pm1.58}$ |
| | Sub-Indep | $66.16_{\pm0.16}$ | $51.13_{\pm0.20}$ | $\mathbf{73.30}_{\pm\mathbf{1.00}}$ | $\mathbf{62.10}_{\pm\mathbf{0.29}}$ | $71.13_{\pm0.33}$ | $59.58_{\pm0.55}$ | $71.37_{\pm0.44}$ | $59.97_{\pm0.41}$ |

setups. The findings are intriguing. The best results among the four methods in the random-label subject-dependent setup are 97.12%, 89.71%, 99.83%, and 72.54% for the ADFTD, TDBrain, PTB, and PTB-XL datasets, respectively, which are unexpectedly high.

Considering that the linkage between disease features and labels are broken after label shuffling, the model's performance should, in theory, be completely random if it relies solely on disease-related features $x^d$ to classify the post-shuffled label $y$. However, taking the Medformer results as an example, the performance drop from the subject-dependent to the random-label subject-dependent setup is almost negligible for both ADFTD and PTB. This indicates that the model learns almost nothing about disease-related features $x^d$ during training in the subject-dependent setup and instead relies entirely on identity features $x^i$ as a shortcut for label $y$ classification. These findings are consistent with the results from the subject-discrimination setup, where ADFTD and PTB also achieved high classification performance, demonstrating their strong identity features. For the TDBrain and PTB-XL datasets, which show relatively weaker identity features in the subject-discrimination setup, the performance drop from subject-dependent to random-label subject-dependent using Medformer is approximately 10% and 17% F1 score, respectively. This indicates that the model does, in fact, learn some disease-related features $x^d$ in the subject-dependent setup for these two datasets.

In summary, these results suggest that, depending on the dataset, it is possible for the model $f_\theta$ to achieve a very high F1 score (over 95%) under the subject-dependent setup while learning little to nothing about the meaningful disease-related features $x^d$ in the dataset. One might assume that increasing the number of subjects could help the model focus more on general disease-related features $x^d$ rather than relying on identity features $x^i$ as shortcuts. However, even with the large PTB-XL dataset containing 17,596 subjects, identity features still had a significant impact on performance. For instance, Medformer's performance in the random-label subject-dependent setup reached an F1 score of 71.15%, which easily surpassed the 59.97% F1 score achieved in the subject-independent setup. Therefore, we strongly recommend that researchers avoid using the subject-dependent setup for disease diagnosis tasks when working with Type-III MedTS datasets.

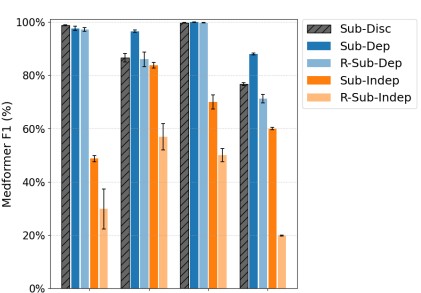

Figure 2: Medformer's F1 (%) across datasets and evaluation protocols. See Table 1 and Section 4 for details in evaluation protocols

Instead, adopting the subject-independent setup provides a more realistic evaluation that aligns with real-world applications.

### 4.4 RESULTS OF RANDOM-LABEL SUBJECT-INDEPENDENT

For completeness, we apply the random-label subject-independent setup to test whether extraneous features $x^o$ significantly affect the model's performance. The results are around chance levels, confirming the irrelevance of $x^o$ in model prediction. Appendix E presents more details.

## 4.5 Results Analysis

We analyze the comparative results and summarize the key observations taking Medformer as an example (Figure 2).

**Identity-based shortcut.** Although label shuffling (R- protocols) typically yields chance-level performance in other contexts, *R-Sub-Dep* attains high F1. This indicates that the model exploits identity features $x^i$ as a shortcut when each subject's label is randomly assigned.

**Identity features outpredict disease features.** The consistent ordering R-Sub-Dep > Sub-Indep implies that identity feature $x^i$ is more salient and predictive than disease features $x^d$ (see Table 1 for feature components used by each protocol). This provides strong evidence that a disease detection model is more prone to utilize $x^i$ than $x^d$ even given true disease labels (Geirhos et al., 2020).

**Identity salience varies by dataset.** The Sub-Dep vs. R-Sub-Dep gap is not constant across datasets and negatively correlates with Sub-Disc performance, indicating that higher Sub-Disc (stronger $x^i$) reduces the added benefit of $x^d$ in Sub-Dep, yielding a smaller gap.

## 5 Discussion

**Insights.** Our findings suggest that the primary factor undermining generalization in Type-III MedTS datasets is the reliance on identity-based shortcuts. This challenge is unique to Type-III settings because subject identity and task label are perfectly correlated, posing a level of difficulty that exceeds standard domain generalization problems. Moreover, the implicit nature of time-series data complicates the separation of identity and task features, since it remains unclear which signal components (e.g., specific frequency bands or temporal patterns) reflect identity rather than disease-related information. As a result, devising data augmentations that eliminate identity features while preserving task features proves non-trivial.

Nonetheless, several promising strategies exist. One avenue is to substantially increase the number of subjects, making identity recognition so difficult that it ceases to function as a reliable shortcut. Achieving this may require large-scale data collection or developing a foundation model trained across multiple datasets. Another direction is to design learning mechanisms that explicitly disentangle task features from identity features (Higgins et al., 2018).

It is also worth emphasizing that simply adopting a more powerful network architecture may be counterproductive: while such a model could learn disease-relevant information more effectively, it might also capture identity cues with greater fidelity. As long as identity features remain dominant, the model's classifier is likely to prioritize identity features over genuine disease-related patterns.

**Limitations and Future Work.** Although our work provides a comprehensive examination of Type-III (multi-subject, single-class) MedTS datasets, it has certain limitations. First, our experiments are confined to Type-III datasets. Since Type-I and Type-II MedTS datasets differ markedly in terms of their subject–label configurations, future research should investigate whether and to what extent identity-based shortcut learning persists in these other categories. Second, while we establish that identity features exist and can heavily influence model predictions, we have not pinpointed exactly what kind of information are contained in identity features (a combination of demographics, physiological and neurological features, perhaps). A more detailed breakdown of these features would be invaluable in guiding effective data augmentation and model design strategies.

## 6 Conclusion

This paper introduces a new taxonomy for MedTS datasets that accounts for both dataset composition and subject dependency, and it further proposes a decomposition of MedTS data into disease features $x^d$, identity features $x^i$, and extraneous features $x^o$. By incorporating new experimental protocols, specifically, *subject-discrimination* and *random-label subject-dependent* setups, we reveal that models heavily leverage on identity features and can achieve deceptively high performance in subject-dependent evaluations while learning minimal disease-related information. These findings highlight the critical need for adopting *subject-independent* evaluation protocols in MedTS research. Looking ahead, developing methods to neutralize or disentangle subject-specific features will be essential for ensuring generalizable, clinically meaningful disease detection.

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

# APPENDIX A  DATA PREPROCESSING

## A.1  TDBRAIN PREPROCESSING

The TDBrain dataset[1], referenced in the paper van Dijk et al. (2022), is a large permission-accessible EEG time series dataset recording brain activities of 1274 subjects with 33 channels. Each subject has two trials: one under eye open and one under eye closed setup. The dataset includes a total of 60 labels, with each subject potentially having multiple labels indicating multiple diseases simultaneously. In this paper, we utilize a subset of this dataset containing 25 subjects with Parkinson's disease and 25 healthy controls, all under the eye-closed task condition. Each eye-closed trial is segmented into non-overlapping 1-second samples with 256 timestamps, and any samples shorter than 1-second are discarded. This process results in 6,240 samples. Each sample is assigned a subject ID to indicate its originating subject. For the training, validation, and test set splits, we employ the subject-independent setup. Samples with subject IDs {18,19,20,21,46,47,48,49} are assigned to the validation set, while samples with subject IDs {22,23,24,25,50,51,52,53} are assigned to the test set. The remaining samples are allocated to the training set.

## A.2  ADFTD PREPROCESSING

The **A**lzheimer's **D**isease and **F**rontotemporal **D**ementia (ADFTD) dataset[2], referenced in the papers Miltiadous et al. (2023b;a), is a public EEG time series dataset with 3 classes, including 36 Alzheimer's disease (AD) patients, 23 Frontotemporal Dementia (FD) patients, and 29 healthy control (HC) subjects. The dataset has 19 channels, and the raw sampling rate is 500Hz. Each subject has a trial, with trial durations of approximately 13.5 minutes for AD subjects (min=5.1, max=21.3), 12 minutes for FTD subjects (min=7.9, max=16.9), and 13.8 minutes for HC subjects (min=12.5, max=16.5). A bandpass filter between 0.5-45Hz is applied to each trial. We downsample each trial to 256Hz and segment them into non-overlapping 1-second samples with 256 timestamps, discarding any samples shorter than 1 second. This process results in 69,752 samples. For the training, validation, and test set splits, we employ both the subject-dependent and subject-independent setups. For the subject-dependent setup, we allocate 60%, 20%, and 20% of total samples into the training, validation, and test sets, respectively. For the subject-independent setup, we allocate 60%, 20%, and 20% of total subjects with their corresponding samples into the training, validation, and test sets, respectively.

## A.3  PTB PREPROCESSING

The PTB dataset[3], referenced in the paper PhysioBank (2000), is a public ECG time series recording from 290 subjects, with 15 channels and a total of 8 labels representing 7 heart diseases and 1 health control. The raw sampling rate is 1000Hz. For this paper, we utilize a subset of 198 subjects, including patients with Myocardial infarction and healthy control subjects. We first downsample the sampling frequency to 250Hz and normalize the ECG signals using standard scalers. Subsequently, we process the data into single heartbeats through several steps. We identify the R-Peak intervals across all channels and remove any outliers. Each heartbeat is then sampled from its R-Peak position, and we ensure all samples have the same length by applying zero padding to shorter samples, with the maximum duration across all channels serving as the reference. This process results in 64,356 samples. For the training, validation, and test set splits, we employ the subject-independent setup. Specifically, we allocate 60%, 20%, and 20% of the total subjects, along with their corresponding samples, into the training, validation, and test sets, respectively.

## A.4  PTB-XL PREPROCESSING

The PTB-XL dataset[4], referenced in the paper Wagner et al. (2020), is a large public ECG time series dataset recorded from 18,869 subjects, with 12 channels and 5 labels representing 4 heart diseases and 1 healthy control category. Each subject may have one or more trials. To ensure consistency, we discard subjects with varying diagnosis results across different trials, resulting in 17,596 subjects

---

[1]https://brainclinics.com/resources/

[2]https://openneuro.org/datasets/ds004504/versions/1.0.6

[3]https://physionet.org/content/ptbdb/1.0.0/

[4]https://physionet.org/content/ptb-xl/1.0.3/

remaining. The raw trials consist of 10-second time intervals, with sampling frequencies of 100Hz and 500Hz versions. For our paper, we utilize the 500Hz version, then we downsample to 250Hz and normalize using standard scalers. Subsequently, each trial is segmented into non-overlapping 1-second samples with 250 timestamps, discarding any samples shorter than 1 second. This process results in 191,400 samples. For the training, validation, and test set splits, we employ the subject-independent setup. Specifically, we allocate 60%, 20%, and 20% of the total subjects, along with their corresponding samples, into the training, validation, and test sets, respectively.

## APPENDIX B    DETAILED EVALUATION PROTOCOLS

**Subject-dependent setup.** Depending on whether a causal split is applied, the subject-dependent setup can be further divided into two subtypes: **Mixed (random split):** The splitting of samples from each subject does not account for causal relationships; instead, samples are randomly mixed (Dai et al., 2019; Sarkar et al., 2022; Seal et al., 2021). This means that stratification occurs without regard for temporal order. **Causal (temporal split):** The split considers the temporal relationship in the time series for each subject, the past samples included in the training set, and future samples placed in the validation and test sets. This causal setup is necessary for time series forecasting tasks (Wu et al., 2021; Zhou et al., 2021; Liu et al., 2024) but is rarely adopted in existing time series classification tasks.

**Subject-independent setup.** This protocol can be further divided into two subtypes: **Leave-One-Out:** Leave only one subject for testing, while all other subjects are used for training (Zhang et al., 2023; Li et al., 2022; Miltiadous et al., 2023a). Due to the limited sample size of a single subject, cross-validation is generally necessary to avoid biased evaluations. However, this setup does not include a dedicated validation set, which can lead to overfitting. **Leave-N-Out:** Leave $N$ subjects for validation and/or test sets (Ahmed et al., 2020; Pandey & Seeja, 2022; Wang et al., 2024c). In this setup, a larger pool of subjects is used for testing, making it more robust than the leave-one-out approach.

## APPENDIX C    IMPLEMENTATION DETAILS

We implement the baselines based on the Time-Series-Library project[5] from Tsinghua University Wu et al. (2022). The four baselines we employ are MLP Popescu et al. (2009), TCN Bai et al. (2018), Transformer Vaswani et al. (2017), and Medformer Wang et al. (2024c).

For MLP, we employ 2 layers with hidden dimension 256. For Transformer and Medformer, we employ 6 layers for the encoder, with the self-attention dimension $D$ set to 128 and the hidden dimension of the feed-forward networks set to 256. The patch list for Medformer is set to {2,4,8} as default. For TCN, we employ 6 layers for the encoder, with hidden dimension 128 and kernel size 3.

## APPENDIX D    DISCUSSION ON TYPE-I MEDTS

In certain MedTS classification tasks, such as personalized health monitoring or brain-computer interface (BCI) systems, where the goal is to train a model that can efficiently control a device (e.g., a wheelchair) for a specific individual, the subject-dependent setup becomes reasonable. In these cases (Type-I MedTS datasets) the training, validation, and test sets are derived from the same subject. This setup aligns with the task's objective of personalizing the model to perform optimally for a single subject, where cross-subject generalization is not the primary focus. We also emphasize the importance of maintaining temporal order in such classification tasks. The evaluation of such Type-I cases is an important future direction.

## APPENDIX E    MORE RESULTS ON RANDOM-LABEL SUBJECT-DEPENDENT

**Setup.** Similar to the random-label subject-dependent setup, we apply the same process in the subject-independent setup, where we randomly assign a new label to each subject while retaining their subject

---

[5]https://github.com/thuml/Time-Series-Library

Table 5: The table presents the results of Type-III datasets evaluated under the random-label subject-independent setup (R-Sub-Indep), along with the results of the original subject-independent setup (Sub-Indep) for comparison. As expected, the performance dropped to random under the random-label subject-independent setup, showing that other features $x_o$ do not impact the model's performance.

| Models | | MLP | | TCN | | Transformer | | Medformer | |
|---|---|---|---|---|---|---|---|---|---|
| Datasets | Setups | Accuracy | F1 Score | Accuracy | F1 Score | Accuracy | F1 Score | Accuracy | F1 Score |
| ADFTD | Sub-Indep | $49.10_{\pm1.01}$ | $43.78_{\pm0.28}$ | $50.46_{\pm1.35}$ | $47.32_{\pm1.27}$ | $50.47_{\pm2.14}$ | $48.09_{\pm1.59}$ | $\mathbf{52.37_{\pm1.51}}$ | $\mathbf{48.72_{\pm1.18}}$ |
| (3-Classes) | R-Sub-Indep | $30.84_{\pm1.99}$ | $29.46_{\pm1.39}$ | $\mathbf{33.25_{\pm6.53}}$ | $\mathbf{31.82_{\pm7.08}}$ | $30.92_{\pm5.25}$ | $29.70_{\pm6.21}$ | $30.67_{\pm7.05}$ | $29.82_{\pm7.53}$ |
| TDBrain | Sub-Indep | $69.42_{\pm0.64}$ | $69.37_{\pm0.64}$ | $83.98_{\pm2.31}$ | $83.93_{\pm2.35}$ | $\mathbf{86.58_{\pm0.76}}$ | $\mathbf{86.52_{\pm0.79}}$ | $83.92_{\pm1.01}$ | $83.69_{\pm1.09}$ |
| (2-Classes) | R-Sub-Indep | $52.48_{\pm6.23}$ | $50.64_{\pm5.77}$ | $55.71_{\pm10.01}$ | $52.55_{\pm9.46}$ | $\mathbf{59.44_{\pm6.60}}$ | $\mathbf{57.21_{\pm6.50}}$ | $58.71_{\pm5.45}$ | $56.82_{\pm4.94}$ |
| PTB | Sub-Indep | $77.76_{\pm0.46}$ | $70.02_{\pm0.60}$ | $\mathbf{83.97_{\pm2.26}}$ | $\mathbf{78.99_{\pm3.44}}$ | $77.37_{\pm1.02}$ | $68.47_{\pm2.19}$ | $77.86_{\pm1.64}$ | $69.93_{\pm2.69}$ |
| (2-Classes) | R-Sub-Indep | $50.27_{\pm6.39}$ | $49.91_{\pm6.34}$ | $\mathbf{50.54_{\pm5.28}}$ | $49.93_{\pm5.45}$ | $48.38_{\pm2.18}$ | $48.16_{\pm1.99}$ | $50.38_{\pm2.58}$ | $\mathbf{49.98_{\pm2.49}}$ |
| PTB-XL | Sub-Indep | $66.16_{\pm0.16}$ | $51.13_{\pm0.20}$ | $\mathbf{73.30_{\pm1.00}}$ | $\mathbf{62.10_{\pm0.29}}$ | $71.13_{\pm0.33}$ | $59.58_{\pm0.55}$ | $71.37_{\pm0.44}$ | $59.97_{\pm0.41}$ |
| (5-Classes) | R-Sub-Indep | $20.00_{\pm0.19}$ | $\mathbf{19.94_{\pm0.16}}$ | $19.76_{\pm0.49}$ | $19.65_{\pm0.55}$ | $\mathbf{20.18_{\pm0.45}}$ | $19.89_{\pm0.39}$ | $19.89_{\pm0.21}$ | $19.79_{\pm0.18}$ |

IDs before splitting the data into training, validation, and test sets. The primary goal of this setup is to eliminate the potential influence of other features $x^o$ on model performance. By masking the disease-related features $x^d$ during training and ensuring that the model $f_\theta$ cannot use subject-specific features $x^s$ as shortcuts under the subject-independent setup during validation and testing, we aim to determine whether any other features $x^o$ besides disease-related and subject-specific features have any impact on model performance. In theory, if the results are completely random under this setup, it would indicate that no other features significantly affect the model's performance.

**Results.** The results of the random-label subject-independent setup are presented in Table 5. As expected, the results for all four datasets are nearly random, corresponding to the number of classes in each dataset. For instance, while some methods, such as the Transformer on the TDBrain dataset, achieved a slightly higher F1 score of 57.21% (compared to the random 50% baseline), this is likely due to the relatively limited number of samples in TDBrain compared to the other three datasets. We consider this slight deviation acceptable. Overall, these results confirm that features other than disease-related and subject-specific features do not affect the model's performance, reinforcing the conclusion that the subject-specific and disease-related features are the primary contributors to classification accuracy in MedTS datasets.

