# OpenReview forum: "Why Do Medical Time Series Models for Disease Detection Generalize Poorly to Unseen Subjects?"
_ICLR.cc/2026/Conference — Submitted to ICLR 2026_

### Official Review · Reviewer_6pps · 2025-10-15

**Soundness:** 3
**Presentation:** 2
**Contribution:** 1
**Rating:** 2
**Confidence:** 4

**Summary:**

The authors consider disease detection from medical time series data, namely ECG and EEG data. The main point of the contribution is to investigate the impact of subject-level vs. sample-level train-test splits, where the latter can lead to an overly optimistic assessment of the model's generalization capabilities to unseen data. The authors formalize the problem and investigate it based on four different datasets (2 ECG, 2 EEG) using different prediction models, demonstrating sizable differences between the two scenarios.

**Strengths:**

* The authors attempt to formalize the problem by categorizing datasets into three categories, based on number of subjects and number of labels and also on the feature side by introducing the notion of disease features, identity features and other features.
* They use a rather comprehensive set of four datasets covering four modalitites and a very comprehensive set of models ranging from MLPs over CNNs to transformer models to demonstrate the robustness of their findings.
* They try to provide further insights into the nature of how identity features are exploited by performing experiments with shuffled labels Random-Label/Subject-Dependent/Independent
* Experiments are presented in a structured way
* The authors quantify the effect

**Weaknesses:**

* The main weakness concerns the topic of the submission itself: The effect of subject-wise vs. sample-wise splits has been recognized years ago and widely accepted best-practice papers such as [1] leveraging PTB-XL clearly advocate for the use of the provided subject-wise splits provided as part of the dataset. The main insight of the paper to use subject-wise splits just reflects a by now common practice and certainly not a major shortcoming in the medical time series community. Results considered as reliable by the community are commonly obtained through subject-wise splits.
* The authors take a very negative point of view towards exploitation of identity features. This is a fair point when considering generalization to unseen subjects, however, neglects the fact some prediction problems are very hard to solve accurately either due to high degree of individualization (e.g. BCI) or due to poor signal to noise ratio (e.g. blood pressure estimation from PPG data). Therefore there certainly is a reason to exist for sample-wise splits, where some samples of a given subjects reside in the training set for calibration purposes and other reside in the test set to assess generalization to "seen" patients.
* In its current form the paper does not make a strong point as it only demonstrates an effect that is already widely known. Nevertheless, I believe that quantifying and characterizing the difference between subject-wise and sample-wise splits is actually a valid contribution, but in scenarios where the latter does not represent a processing error but a valid application scenario (BCI with/without calibration, BP estimation with/without calibration). I don't think that ECG is a good example to make such a case.
* I am not convinced that the chosen metrics are really suitable to characterize the overall discriminative power of the models, consider include other metrics such as AUROC, logloss etc. It would also be helpful to align with metrics that are used in the literature for the respective datasets.
* The mathematical introduction of the feature decomposition in 3.1 seems rather ad-hoc and not well-justified. For the sake of the argument they could just leave it out and argue more qualitatively.

[1] Strodthoff, N., Wagner, P., Schaeffter, T., & Samek, W. (2020). Deep learning for ECG analysis: Benchmarks and insights from PTB-XL. IEEE journal of biomedical and health informatics, 25(5), 1519-1528.

**Questions:**

How does the dataset categorization seems to be very much focused on binary labels (as are also the experiments, which artificially enforce single labels). How does it align with multi-label datasets as commonly encountered in medical applications?

---

### Official Review · Reviewer_bfLu · 2025-10-31

**Soundness:** 3
**Presentation:** 2
**Contribution:** 2
**Rating:** 2
**Confidence:** 5

**Summary:**

This paper investigates why medical time-series (MedTS) models—particularly those used for disease detection from EEG and ECG—often fail to generalize across patients. The authors show that many datasets assign a single label per subject, leading models to exploit *identity-related features* rather than true *disease-related features*. Through systematic experiments across multiple datasets (ADFTD, TDBrain, PTB, PTB-XL) and five evaluation setups, the paper demonstrates that strong performance under subject-dependent evaluation largely stems from patient-specific information rather than generalizable disease features. The work emphasizes the importance of subject-independent evaluation for reliable clinical generalization.

**Strengths:**

1. **Fundamental Contribution to Clinical ML Evaluation**
   The paper identifies and formalizes a foundational issue that extends beyond time-series analysis: **patient-level data leakage**. In many medical AI studies—including ECG, EEG, and imaging—evaluation protocols fail to separate subjects across splits, inflating perceived generalization. This work systematically exposes how such leakage manifests in time-series data and provides strong empirical evidence for its impact.

2. **Systematic and Comprehensive Experimental Validation**
   The experiments span multiple datasets (PTB, PTB-XL, ADFTD, TDBrain) and five evaluation configurations, offering a convincing and generalizable demonstration of the identity leakage phenomenon. The consistent trend across datasets reinforces the central argument.

**Weaknesses:**

1. **Interpretation of R_sub_dep vs. sub_ind Results**
   The paper claims that higher performance in the random subject-dependent (R_sub_dep) setup proves models rely on identity features. However, this interpretation is not fully substantiated. The observed gap may also arise from **data-level leakage** introduced by preprocessing. Specifically, PTB-XL’s 10-second ECGs are divided into ten 1-second segments, each treated as an independent sample. Given ECG’s periodicity, maybe these adjacent segments are semantically identical, meaning training and test samples may share nearly identical signal content. This could artificially inflate R_sub_dep performance even without explicit identity learning. The authors should verify that this segmentation strategy does not cause intra-patient leakage and clarify whether the R_sub_dep vs. sub_ind difference truly reflects identity exploitation rather than temporal redundancy.

2. **Ambiguity in Table 4 Presentation**
   The use of boldface to highlight the highest score in each row is misleading. Since the experiment aims to compare **evaluation setups**, not architectures, bolding the best model per row implies architectural superiority unrelated to the paper’s objective. The authors should remove or clarify the use of bold formatting.

3. **Preprocessing Transparency and Segment Correlation**
   The reduction in subject count for PTB-XL suggests that samples with inconsistent labels were excluded, but this is undocumented.

4. **Random-Label Experimental Design**
   The random-label setup is conceptually useful but under-specified. It is unclear whether class distributions were preserved during randomization. Without maintaining label distribution, performance drops might reflect sampling bias rather than absence of identity features.

5. **Limited Analytical Depth on Identity Features**
   While the study confirms the presence of identity-related effects, it stops short of exploring *what* those features represent (e.g., age, sex, acquisition protocol, electrode characteristics). Investigating this aspect could significantly strengthen the paper’s interpretability and impact.

**Questions:**

1. How do you ensure that the observed performance gap between R_sub_dep and sub_ind truly reflects identity learning?

2. Were any steps taken to prevent adjacent 1-second ECG segments from being distributed across train and test splits? If not, could this explain part of the R_sub_dep advantage?

3. Why are some R_sub_ind results missing from the reported tables, despite being central to the paper’s argument?

4. In the random-label experiment, were class distributions preserved to avoid bias from label imbalance?

---

### Official Review · Reviewer_9vkw · 2025-10-31

**Soundness:** 2
**Presentation:** 3
**Contribution:** 2
**Rating:** 2
**Confidence:** 4

**Summary:**

This paper investigates why medical time-series (MedTS) models fail to generalize to unseen subjects. The authors attribute this issue to identity features, where models exploit subject-specific information rather than disease-relevant features. The paper introduces a taxonomy of MedTS datasets (Type-I, II, III) and proposes several evaluation setups to reveal how identity cues drive performance.

**Strengths:**

1) The taxonomy of MedTS dataset types (Type-I/II/III) and evaluation setups (Sub-Dep, Sub-Indep, Sub-Disc, R-Sub-Dep, R-Sub-Indep) provides a structured framework for analysis of MedTS data.
2) The “random-label” and “subject-discrimination” experiments are good diagnostic tools that clearly expose identity feature dominance.
3) The drop in F1-scores under subject-independent evaluation compared to random-label subject-dependent supports the paper’s central idea.

**Weaknesses:**

1) The experiments are conducted exclusively on Type-III MedTS datasets. Although the authors acknowledge this limitation, analyzing only Type-III data does not fully justify the paper’s rather general title.
2) Terms like “identity features” and “disease features” are used intuitively but not operationally defined (e.g., which channels, frequencies, or components contribute). A feature attribution analysis could improve interpretability as well as make the claims stronger.
3) The paper is clear overall but verbose in places; tightening sections 3 and 4 would improve readability.

**Questions:**

1) The average number of images per subject for PTB-XL dataset is $\frac{191400}{17596}$ \~ $10$. In the subject-discrimination setup, does this limited per-subject data risk overfitting the model to small sample artifacts instead of genuine subject identity features? How do the authors mitigate or validate against this possibility?

The authors are requested to address the identified weaknesses of the paper and provide responses to them.

---

### Meta-Review · Area_Chair_zVhz · 2025-12-23

**Summary:**

Summary of contributions:
The paper proposes simple tests to identify whether medical time-series models rely on sample identity features over disease-related features for classification. This tendency of medical time-series models has been widely noted but not thorougly studied. The paper proposes three tests to study in conjunction i) subject discrimation: to assess the existence of identity/subject-identifying features, if accuracy is high, it indicates existence. ii) random-label subject dependent tests compared to subject dependent tests: most often, models use a combination of disease-specific features and identity features to predict a disease label. If a disease label is randomized but consistent across subject-specific samples, then the model will rely on identity features to predict the disease label. Comparing that to a model trained to predict disease label directly (which uses a combination of disease and identity features) can reveal the dependence on identity features. That is if the gap is high, and the drop in performance for random-label subject dependent data is significant, then the model uses disease specific labels over identity labels. If the drop is insignificant or model is highly performant, indicates dependence on identity features. It is possible that model relies on a third set of features, to control for which authors propose random-label prediction that also breaks subject-level association.

On 4-6 model classes and four datasets, it is clear that medical timeseries models tend to contain identity-related artefacts, thend to rely on identity features for disease classification, but do so with varying degrees. This suggest the fragility of medical time series models.

Reviewer comments and concerns:

Strengths:
   1. Reviewers commented on formalization and taxonomy of evaluation of medical time-series to be useful.
   2. Reviewers identified that the operationalizing these into specific testable experiments is well done, and support the initial hypothesis
   3. Reviewers considered the evaluation comprehensive.

Weakness:
1. Reviewers pointed out that some of the evaluation may be underspecified and evaluations of the randomized label subject-dependent tests do not rule out other potential hypotheses like leakage and may have been overattributed to learning on identity features.
2. Lack of good operationalization of identity vs disease-specific features
3. The choice of metrics and/or dataset may not be sufficient to confirm the issue the paper is trying to highlight.
4. The challenges is well known and characterized which minimizes the impact of the contribution
5. Readability and presentation are poor and could be significantly tightened.

**Reviewer Concerns:**

None of the reviewer concerns were addressed. A rebuttal was not submitted

**Reviewer Scores:**

No rebuttal received so I expect the scores to be the same.

---

### Decision · Program_Chairs · 2026-01-26

Reject